# Dorsal–Ventral Gradient of Activin Regulates Strength of GABAergic Inhibition along Longitudinal Axis of Mouse Hippocampus in an Activity-Dependent Fashion

**DOI:** 10.3390/ijms241713145

**Published:** 2023-08-24

**Authors:** Maria Jesus Valero-Aracama, Fang Zheng, Christian Alzheimer

**Affiliations:** Institute of Physiology and Pathophysiology, Friedrich-Alexander-Universität Erlangen-Nürnberg, 91054 Erlangen, Germany

**Keywords:** activin, hippocampus, CA1 pyramidal cell, tonic GABA current, GABA_A_ receptors, inhibitory postsynaptic current, environmental enrichment

## Abstract

The functional and neurophysiological distinction between the dorsal and ventral hippocampus affects also GABAergic inhibition. In line with this notion, ventral CA1 pyramidal cells displayed a more dynamic and effective response to inhibitory input compared to their dorsal counterparts. We posit that this difference is effected by the dorsal–ventral gradient of activin A, a member of the transforming growth factor-β family, which is increasingly recognized for its modulatory role in brain regions involved in cognitive functions and affective behavior. Lending credence to this hypothesis, we found that in slices from transgenic mice expressing a dominant-negative mutant of activin receptor IB (dnActRIB), inhibitory transmission was enhanced only in CA1 neurons of the dorsal hippocampus, where the basal activin A level is much higher than in the ventral hippocampus. We next asked how a rise in endogenous activin A would affect GABAergic inhibition along the longitudinal axis of the hippocampus. We performed ex vivo recordings in wild-type and dnActRIB mice after overnight exposure to an enriched environment (EE), which engenders a robust increase in activin A levels in both dorsal and ventral hippocampi. Compared to control mice from standard cages, the behaviorally induced surge in activin A produced a decline in ventral inhibition, an effect that was absent in slices from dnActRIB mice. Underscoring the essential role of activin in the EE-associated modulation of ventral inhibition, this effect was mimicked by acute application of recombinant activin A in control slices. In summary, both genetic and behavioral manipulations of activin receptor signaling affected the dorsal–ventral difference in synaptic inhibition, suggesting that activin A regulates the strength of GABAergic inhibition in a region-specific fashion.

## 1. Introduction

Activins belong to the transforming growth factor β (TGF-β) family. They act as multifunctional regulatory proteins in many organ systems including the brain [1,2]. Activin A, a homodimer of two βA subunits, is the most abundant and relevant activin variant in the brain, with proven roles in neural development [3], neuroprotection [4,5,6], and neural plasticity in regions involved in learning and memory [7,8,9,10]. Activin A levels are strongly responsive to various forms of enhanced neuronal activity, ranging from brief stimulus trains that induce long-term potentiation of excitatory synapses [3,11] to overnight exploration of an enriched environment (EE) [12]. Activin levels rise in a graded, stimulation-dependent fashion, such that, for example, overnight exposure to EE produces a moderate, but robust and distributed, pattern of enhanced activin A in the mouse hippocampus, whereas delivery of an electroconvulsive therapy (ECT)-like regimen engenders a tremendous surge of activin A that is restricted to the dentate gyrus [12]. Putting the previous findings in a nutshell, activin A is likely to assume a dual role in the adult brain, acting as a “master molecule” that tunes the neural underpinnings of cognitive and affective processing under physiological conditions, while serving as a promising therapeutic target in various neuropsychiatric disorders [8,9,13,14,15,16,17,18,19].

In addition to its well-characterized effects at glutamatergic synapses, activin signaling has a sizeable impact on the GABA system and related behavior [14,15,20]. In hippocampal slices from transgenic mice with a dominant-negative mutant of activin receptor IB (dnActRIB) that disrupts activin signaling in forebrain neurons, we found significant changes in tonic and phasic GABAergic inhibition, in allosteric modulation of GABA_A_ receptors (GABA_A_Rs) by benzodiazepines and ethanol, and in GABA_B_ receptor-mediated K^+^ current [20,21,22]. Furthermore, overexpression of activin or its functional antagonist follistatin produced opposite effects on anxiety-related behavior [23].

So far, the mechanisms of activin signaling in cognitive function and affective behavior have been mainly studied in the dorsal part of the hippocampus (DH). However, for a comprehensive understanding of activin signaling, it is also essential to elucidate its effects in the ventral part of the hippocampus (VH), given that these two regions are endowed with distinct anatomical and functional features. Macroscopically, the rodent hippocampus presents as an elongated structure with a longitudinal axis extending in a C-shape fashion from dorsal to ventral, corresponding to posterior to anterior in humans [24]. Accumulating evidence supports a functional segregation along the longitudinal axis, with DH preferentially associated with cognitive functions, in particular spatial information processing, and VH predominantly involved in emotional processing and affective behavior [25,26]. In addition to their distinct cortical/subcortical connectivity, pyramidal cells from DH and VH differ in their basal electrophysiological properties, in their set of ion channels, and in their synaptic properties [27,28,29,30,31]. In rodent models of epilepsy, DH and VH vary in the degree of seizure susceptibility [32,33]. Finally, at the molecular level, a strong dorsal–ventral asymmetry in transcription and epigenetic regulation was reported between the dorsal and ventral dentate gyrus of adolescent mice, with the difference becoming even more pronounced after environmental enrichment [34].

Considering the preferential involvement of VH in emotional responses as well as in epileptic seizures, we know surprisingly little about the specifics of GABAergic inhibition in this part of the hippocampus. Milior et al. [29] found a higher frequency of miniature GABAergic events in ventral than in dorsal mouse CA1 pyramidal cells, which would agree with the earlier finding that the level of extracellular GABA is higher in VH than in DH [35]. With respect to GABA_A_ receptor density, DH prevails over VH, although receptor subtypes differ between the regions [36,37].

In view of the incomplete knowledge on the ventral GABA system and its regulation by activin signaling, we undertook a comprehensive assessment of the tonic and phasic components of GABA_A_R-mediated inhibition in DH vs. VH, elucidated the contribution of basal activin signaling to region-specific differences in inhibition, and explored how a behaviorally induced rise in activin such as that provided by environmental enrichment redistributes the weight of synaptic inhibition between DH and VH.

## 2. Results

### 2.1. Activin A Levels Exhibit Dorsal–Ventral Gradient, Which Is Strongly Potentiated by Enriched Environment

We have previously shown that overnight exploration of an EE strongly upregulates expression of *Inhba* mRNA (which encodes the activin βA subunit) in the mouse hippocampus [12], leading to a subsequent surge in activin A, a dimeric protein composed of two βA subunits [10,22]. To determine whether EE boosts activin A levels differentially along the hippocampal longitudinal axis in control wild-type (wt) mice, we trimmed hippocampi into three parts of about equal weight, namely dorsal hippocampus (DH, control: 8.93 ± 0.30 mg, n = 9; EE: 8.43 ± 0.24 mg, n = 8), middle hippocampus (MH, control: 10.68 ± 0.43 mg; EE: 10.44 ± 0.48 mg), and ventral hippocampus (VH, control: 10.34 ± 0.39 mg; EE: 9.96 ± 0.45 mg). As shown in Figure 1A, basal levels of activin A gradually declined from DH to VH (n = 9; DH, 2.99 ± 0.17 pg/mg; MH, 1.29 ± 0.10 pg/mg; VH, 0.84 ± 0.07 pg/mg; one-way AVOVA followed by Tukey’s post hoc test: F(1,8) = 51.28, *p* =1.53 × 10^−8^; DH vs. MH *p* = 2.01 × 10^−7^, DH vs. VH *p* = 2.42 × 10^−9^, MH vs. VH *p* = 0.002). Overnight, EE strongly enhanced activin A in all three regions of the hippocampus (n = 8; DH-EE, 7.42 ± 0.20 pg/mg, *p* = 3.51 × 10^−11^ vs. DH-control; MH-EE, 4.20 ± 0.24 pg/mg, *p* = 8.35 × 10^−9^ vs. MH-control; VH-EE, 2.70 ± 0.21 pg/mg, *p* = 1.64 × 10^−7^ vs. VH-control; two-way AVOVA, F(1,50) = 102.93, *p* = 9.86 × 10^−14^). Although the absolute protein level remained much lower ventrally (*p* = 2.06 × 10^−10^, VH-EE vs. DH-EE), the relative increase after EE (normalized to control) appeared higher in VH than in DH (3.21-fold vs. 2.48-fold). Hippocampi from dnActRIB mice did not differ in weight from those of wt mice (wt-control, n = 9, 30.38 ± 0.87 mg; dnActRIB-control, n = 11, 29.56 ± 1.14 mg, *p* = 0.592), nor did mutant hippocampi differ from their wt counterparts with respect to the extent of the dorsal–ventral gradient of activin A before and after EE (dnActRIB-EE, n = 11; Figure 1B).

### 2.2. Loss of Activin Signaling Enhances Tonic GABA Inhibition in DH > VH

Activation of GABA_A_Rs can exert two temporally and functionally distinct effects on neuronal excitability, commonly referred to as phasic and tonic inhibition [38,39]. The phasic component results from the brief pulse of action potential-dependent or -independent GABA release and is mediated by synaptic GABA_A_Rs, whereas tonic inhibition is caused by low ambient GABA acting predominantly on extrasynaptic GABA_A_Rs. To accentuate the dorsal–ventral distinction in view of the gradual shift of activin A levels along the longitudinal axis, we used in the following only slices from either the dorsal or ventral pole of the hippocampus. First, we examined the extent of tonic inhibition in control mice, as indicated by the shift of holding current in the presence of the GABA_A_R antagonist picrotoxin (PTX, 100 µM; Figure 2A). Voltage-clamped CA1 pyramidal cells (V_h_ −70 mV) displayed a larger tonic current in VH (21.19 ± 2.00 pA, n = 16) than in DH (14.81 ± 1.45 pA, n = 11; *p* = 0.027, Figure 2B, left). Disruption of activin receptor signaling almost doubled tonic inhibition of dorsal CA1 neurons (dnActRIB-DH, n = 6, 24.33 ± 1.45 pA; *p* = 7.32 × 10^−4^ vs. wt-DH), confirming an earlier finding in dorsal CA1, then made at room temperature and in the absence of low exogenous GABA [20]. As expected from the dorsal–ventral gradient of basal activin levels (Figure 1), loss of activin signaling barely changed tonic inhibition in ventral CA1 neurons (dnActRIB-VH, n = 12, 21.87 ± 2.50 pA, *p* = 0.830 vs. wt-VH). As a consequence, the dorsal–ventral difference in GABA tone vanished (*p* = 0.285, Figure 2B, right).

Interestingly, overnight EE housing affected the strength of tonic inhibition only in ventral CA1 pyramidal cells (ventral wt-EE: n = 7, 14.27 ± 0.72 pA, *p* = 0.037 vs. ventral wt-control; Figure 2C, left) and not in dorsal ones (dorsal wt-EE: n = 7, 13.25 ± 2.50 pA, *p* = 0.578 vs. dorsal wt-control; Figure 2C, right). In slices from dnActRIB mice, preceding EE altered tonic inhibition neither in ventral nor in dorsal CA1 pyramidal cells (ventral dnActRIB-EE: n = 6, 23.72 ± 2.32 pA, *p* = 0.525 vs. ventral dnActRIB-control; dorsal dnActRIB-EE: n = 6, 25.34 ± 4.73 pA, *p* = 0.764 vs. dorsal dnActRIB-control; Figure 2C). Taken together, these data strongly suggest that the level of activin A regulates the strength of tonic GABA current in a region-specific fashion.

### 2.3. Disruption of Activin Signaling Facilitates Phasic Inhibition Mainly in Dorsal CA1 Pyramidal Cells

We then compared phasic inhibition between CA1 neurons from DH and VH by monitoring spontaneously occurring inhibitory postsynaptic currents (spIPSCs). These events did not display an appreciable bias towards one hippocampal region in CA1 cells from control mice (Figure 3; Table 1). spIPSC frequencies were similar in dorsal and ventral regions (wt-VH, n = 63, 6.26 ± 0.38 Hz; wt-DH, n = 42, 5.77 ± 0.40 Hz; *p* = 0.350), as were their averaged peak amplitudes (wt-VH, 50.27 ± 1.63 pA; wt-DH, 49.35 ± 2.00 pA; *p* = 0.723). It is worth noting, though, that spIPSCs in ventral pyramidal cells had faster rise times (measured from 10–90% of peak: wt-VH, 0.48 ± 0.01 ms; wt-DH, 0.53 ± 0.01 ms; *p* = 0.004), whereas decay tau did not differ (wt-VH, 8.72 ± 0.21 ms; wt-DH 8.26 ± 0.26 ms; *p* = 0.169) (Figure 3; Table 1).

Substantiating our earlier finding made at room temperature [20], dnActRIB CA1 pyramidal cells from DH exhibited more spontaneous events than their wt counterparts (dnActRIB-DH, n = 27, 7.19 ± 0.60 Hz; *p* = 0.039 vs. wt-DH; Figure 3C, top left). However, this increase was not observed in the ventral region, where spIPSC frequency in dnActRIB pyramidal cells (dnActRIB-VH, n = 53, 5.86 ± 0.35 Hz) was not different from that of their wt counterparts (*p* = 0.439; Figure 3B, top left). The unilateral increase in spIPSC frequency in the dorsal part of the mutant hippocampus attained significance (*p* = 0.044) and was associated with a slower decay of the synaptic events (*p* = 0.007) (Figure 3; Table 1). Peak amplitude of spIPSC was similar between wt and dnActRIB of ventral (dnActRIB-VH, n = 53, 51.19 ± 2.14 pA; *p* = 0.728 vs. wt-VH) and dorsal regions (dnActRIB-DH, n = 27, 47.74 ± 2.58 pA; *p* = 0.620 vs. wt-DH) (Figure 3B,C, top right).

### 2.4. EE-Induced Activin Signaling Selectively Attenuates Phasic Inhibition in Ventral Hippocampus

After overnight EE housing of wt mice, spIPSC frequency was significantly reduced in ventral CA1 pyramidal cells (wt-VH-EE: n = 51, 4.41 ± 0.27 Hz; *p* < 0.001 vs. wt-VH; Figure 4A,B, top left), but not in dorsal pyramidal cells (wt-DH-EE, n = 19, 5.81 ± 0.47 Hz; *p* = 0.714 vs. wt-DH; Figure 4C, top left, wt bars). Peak amplitude of spIPSCs remained unchanged after EE (wt-VH-EE: 53.06 ± 2.37 pA, *p* = 0.320 vs. wt-VH; wt-DH-EE: 49.02 ± 2.52 pA, *p* = 0.922 vs. wt-DH; Figure 4B,C, top left, wt bars). EE did not significantly alter ventral spIPSC kinetics (rise time: wt-VH-EE, 0.45 ± 0.01 ms, *p* = 0.070, vs. wt-VH, Figure 4B bottom left, wt bars; decay tau: wt-VH-EE, 8.17 ± 0.19 ms, *p* = 0.057, vs. wt-VH, Figure 4B bottom right, wt bars). In dorsal cells, EE reduced rise time (wt-DH-EE, n = 19, 0.47 ± 0.02 ms, *p* = 0.014 vs. wt-DH, Figure 4C, bottom left, wt bars) but did not change decay tau (wt-DH-EE, 8.67 ± 0.34 ms, *p* = 0.828 vs. wt-DH, Figure 4C bottom right, wt bars) (Table 1).

We next asked whether the selective dampening of GABA inhibition in VH is causally linked to the strong increase in activin signaling during EE. If so, the effect should be abrogated in EE-exposed dnActRIB mice and mimicked in VH slices from wt mice superfused with recombinant activin A. EE indeed failed to affect spIPSC frequency in ventral CA1 pyramidal cells from mutant mice (dnActRIB-VH-EE, n = 25, 5.53 ± 0.48 Hz; *p* = 0.587, vs. dnActRIB-VH; Figure 4B, top left, dnActRIB bars). As expected, EE did not alter spIPSC amplitude in ventral cells of dnActRIB mice (amplitude: dnActRIB-VH-EE: 55.67 ± 2.62 pA, *p* = 0.217 vs. dnActRIB-VH, Figure 4B, top right, dnActRIB bars), nor did we observe a change of these parameters after EE in mutant DH (frequency: dnActRIB-DH-EE: n = 11, 6.28 ± 0.57 Hz, *p* = 0.378 vs. dnActRIB-DH; amplitude: dnActRIB-DH-EE: 51.13 ± 3.07 Hz, *p* = 0.456 vs. dnActRIB-DH; Figure 4C, top, dnActRIB bars). As in the wt preparation, EE did not affect spIPSC kinetics in mutant VH (dnActRIB-VH-EE: rise time 0.48 ± 0.02 ms; decay tau: dnActRIB-VH-EE: 8.56 ± 0.28 ms, *p* = 2.11 vs. dnActRIB-VH-control). Likewise, the reduction of rise time after EE was preserved in mutant DH (0.47 ± 0.02 ms, *p* = 0.018, Figure 4C, bottom left), whereas decay was slower than in DH from mutant mice in standard housing (dnActRIB-DH-EE: 9.56 ± 0.61 ms, *p* = 0.006 vs. dnActRIB-DH-control) (Figure 4B,C, bottom right; Table 1).

To further tighten the link between EE, activin, and phasic inhibition, we superfused hippocampal slices from wt mice with recombinant activin A (25 ng/mL). Administration of the factor for 5–10 min reduced spIPSC frequency in VH pyramidal cells (Figure 5B, spIPSC bars) from 5.63 ± 0.728 Hz to 4.54 ± 0.62 Hz (n = 9, paired *t*-test, *p* = 0.018; Figure 5B, top left). Activin A did not alter peak amplitude of spIPSC (wt-VH, 54.86 ± 4.02 pA; wt-VH-activin A: 56.38 ± 4.25 pA; n = 9; *p* = 0.538, paired *t*-test, Figure 5B, top right). Also, spIPSC kinetics remained unchanged (10–90% rise time: control 0.45 ± 0.04 ms; activin 0.45 ± 0.04 ms; *p* = 0.766, paired *t*-test; decay tau: control 9.54 ± 0.53 ms; activin: 9.71 ± 0.51 ms; *p* = 0.538, paired *t*-test, Figure 5B, bottom). Unlike EE, recombinant activin A decreased spIPSC frequency in dorsal cells, too (Figure 5C, spIPSC bars, n = 9; control: 5.38 ± 0.54 Hz; activin: 5.00 ± 0.50 Hz; *p* = 0.034, paired *t*-test; Figure 5C, top left). Dorsal spIPSC amplitude and rise time did not change during activin perfusion, whereas decay tau was significantly prolonged (wt-DH, 7.48 ± 0.59 ms; activin: 7.91 ± 0.67 ms, *p* = 0.046). Although application of recombinant activin A reduced spIPSC frequency in both hippocampal parts, the relative effect seemed stronger in VH (17.64 ± 4.68%) than in DH (6.59 ± 2.82%, *p* = 0.060). In fact, we had to double the concentration of recombinant activin A (50 ng/mL) to obtain a reduction of dorsal spIPSC frequency similar to that seen in VH at half the concentration (n = 6; from 4.72 ± 0.99 Hz to 4.01 ± 0.91 Hz; *p* = 0.010, paired *t*-test; relative reduction: 14.83 ± 4.54%). The notion of a VH > DH ratio in the sensitivity of GABAergic inhibition to a rise in activin level is in agreement with our above results from the EE paradigm, where only VH was affected. With all the necessary caveats, the slice experiments with the recombinant protein might be taken to extrapolate on the level of endogenous activin A in hippocampal tissue that is attained with EE.

To isolate the effects of activin at inhibitory synapses from putative actions of the protein on firing properties of GABA interneurons, we applied tetrodotoxin (TTX; 0.5 µM) to remove action potential-driven inhibitory events. Recombinant activin A (25 ng/mL for 5–10 min) decreased the frequency of miniature IPSCs (mIPSCs; Figure 5A–C) in both dorsal (n = 7, from 4.32 ± 0.61 Hz to 3.38 ± 0.52 Hz, paired *t*-test, *p* = 0.006) and ventral pyramidal cells (n = 8, from 4.12 ± 0.88 Hz to 3.68 ± 0.81 Hz, paired *t*-test, *p* = 0.026; Figure 5B,C, top left). As illustrated in Figure 5A, the activin A-induced reduction in mIPSC frequency was reversible upon wash-out. Like spIPSCs, activin A did not alter averaged peak amplitude of mIPSCs (dorsal cells, from 43.42 ± 4.53 pA to 41.31 ± 4.11 pA, *p* = 0.202; ventral cells, from 45.49 ± 2.32 pA to 44.87 ± 2.28 pA, *p* = 0.685). Whereas rise time remained unchanged (dorsal cells, from 0.54 ± 0.02 ms to 0.56 ± 0.02 ms, *p* = 0.141; ventral cells, from 0.57 ± 0.02 ms to 0.57 ± 0.03 ms, *p* = 0.640), activin A slowed mIPSC decay in both dorsal and ventral cells (dorsal cells, decay tau from 6.99 ± 0.62 ms to 7.49 ± 0.62 ms, *p* = 0.007; ventral cells, from 7.31 ± 0.44 ms to 7.90 ± 0.62 ms, *p* = 0.042).

### 2.5. Properties of Evoked IPSCs along Longitudinal Axis of Hippocampus

We next compared the properties of electrically evoked IPSCs (eIPSCs) between DH and VH, using a bipolar stimulating electrode placed in the stratum radiatum next to the CA1 pyramidal cell layer. Input–output (I-O) relationships were determined by plotting peak eIPSC amplitude as a function of stimulus intensity (50–200 µA, duration of 0.1 ms). As depicted in Figure 6A, I-O curves for dorsal and ventral CA1 pyramidal cells from wt mice diverged with increasing stimulus intensity, yielding stronger synaptic responses in the latter. This data set was compared using a two-way repeated measures ANOVA, where the within-subject factor is stimulation intensity and the between-subject factor the hippocampal region, resulting in a between-subject difference with F = 5.28 and *p* = 0.030 *(*Figure 6A). At the highest stimulus intensity (200 µA), peak eIPSC amplitude in ventral pyramidal cells (n = 18, 3.16 ± 0.33 nA) was significantly higher than in dorsal cells (n = 11, 1.91 ± 0.20 nA, *p* = 0.011). Dorsal and ventral eIPSCs also differed in their kinetics (Table 2). When determined at 200 µA, ventral responses rose and decayed faster than dorsal ones (10–90% rise time: ventral 2.42 ± 0.14 ms; dorsal 3.65 ± 0.34 ms, *p* < 0.001; decay tau: ventral 47.12 ± 3.77 ms; dorsal 69.61 ± 6.83 ms, *p* = 0.006; Figure 6B). In terms of synaptic efficacy, the slower kinetics of dorsal eIPSCs made up for their lower peak amplitude, yielding very similar values of total charge transfer for dorsal and ventral responses (wt-VH: 0.15 ± 0.01 nA*ms; wt-DH: 0.15 ± 0.02 nA*ms, *p* = 0.903; Figure 6B). It follows that when eIPSCs of about equal peak amplitude (1.2 nA) were compared, synaptic charge transfer was significantly greater in dorsal than in ventral cells (wt-VH: 0.064 ± 0.006 nA*ms; wt-DH: 0.102 ± 0.010 nA*ms, *p* = 0.004).

Upon repetitive stimulation, hippocampal eIPSCs typically exhibit a decline in subsequent responses, a phenomenon termed frequency-dependent depression. When we delivered stimulus trains of 15 pulses at 5 Hz, we found that frequency-dependent depression of eIPSCs differed significantly between dorsal and ventral CA1 pyramidal cells (Figure 6C). In this set of experiments, stimulus intensity was adjusted individually to elicit submaximal (60–80%) eIPSCs with peak amplitudes of 1.0–1.5 nA (1st IPSC: wt-VH, n = 19, 1.55 ± 0.14 nA; wt-DH, n = 11, 1.21 ± 0.11 nA; *p* = 0.100). eIPSC amplitudes were then normalized to that of the first response. As depicted in Figure 6D, frequency-dependent depression was more pronounced in dorsal than ventral CA1 neurons (two-way repeated measures ANOVA, within-subject factor of stimulation number and between-subject factor of anatomical region, F = 14.48, *p* < 0.001). The ratio of second to first synaptic response, the so-called paired-pulse ratio, was 0.69 ± 0.04 in ventral cells (n = 19) and 0.53 ± 0.04 in dorsal cells (n = 11; *p* = 0.005), and the ratio at the end of train (fifteenth to first eIPSC) was 0.61 ± 0.04 in ventral cells (n = 19) and 0.45 ± 0.04 in dorsal cells (n = 11; *p* = 0.019). Again, total charge transfer during repetitive stimulation did not significantly vary between VH and DH, owing to the inverse coupling of kinetics and amplitude, as noted above for single eIPSCs (wt-VH, 0.58 ± 0.01 nA*ms; wt-DH, 0.54 ± 0.01 nA*ms, *p* = 0.747). The same was true when we calculated the relative charge transfer from the normalized responses of Figure 6D (wt-VH; 386.38 ± 28.52; wt-DH = 426.43 ± 31.14, *p* = 0.365, Figure 6E).

### 2.6. Activin Signaling Redistributes Weight of eIPSCs between DH and VH

We next examined how activin signaling shapes eIPSCs in the dorsal and ventral hippocampus. Genetic disruption of activin receptor signaling produced seemingly opposing effects on the I-O relationship in ventral vs. dorsal CA1 pyramidal cells (Figure 7A,B). In mutant VH, we observed an apparent downward shift of the I-O curve that remained, however, below the level of statistical significance (n = 16; two-way repeated measures ANOVA, F = 3.09, *p* = 0.088; Figure 7A), whereas the I-O curve from mutant DH exhibited a robust upward shift (n = 14; two-way ANOVA repeated measures, F = 4.59, *p* = 0.043; Figure 7B). When compared to the respective responses of their wt counterparts at maximum stimulus intensity (200 µA), eIPSC amplitudes in mutant ventral cells were not significantly different (2.34 ± 0.28 nA; *p* = 0.074 vs. wt-VH), whereas mutant dorsal cells did show a significant increase (2.73 ± 0.22 nA; *p* = 0.014 vs. wt-DH). In ventral cells, eIPSC rise time tended to be longer in dnActRIB than in wt neurons (dnActRIB-VH: n = 16, 3.09 ± 0.31 ms, *p* = 0.051 vs. wt-VH, Figure 7C), while decay tau remained in the same range (dnActRIB-VH: 50.33 ± 4.85 ms, *p* = 0.601 vs. wt-VH, Figure 7D) (Table 2). Vice versa, in DH, decay tau was significantly faster in dnActRIB cells (47.29 ± 6.11 ms, *p* = 0.029 vs. wt-DH, Figure 7D), while rise time was unaltered (3.24 ± 0.33 ms, *p* = 0.398 vs. wt-DH, Figure 7C).

Interestingly, the marked dorsal–ventral difference in eIPSC kinetics of wt neurons was equalized in dnActRIB neurons (*p* = 0.740, dorsal–ventral comparison of rise time; *p* = 0.697, dorsal–ventral comparison of decay tau) (Table 2). With respect to frequency-dependent depression, loss of activin receptor signaling entailed a unilateral effect restricted to DH, where depression was significantly attenuated (Figure 7E,F; ventral dnActRIB cells, n = 11, two-way repeated measures ANOVA, F = 0.03 *p* = 0.859; dorsal dnActRIB cells n = 8, two-way repeated measures ANOVA, F = 12.42, *p* = 0.003). The relative charge transfer during the stimulus train was not affected by the mutation, neither in ventral (314.74 ± 30.22, *p* = 0.125 vs. wt-VH) nor in dorsal hippocampus (374.85 ± 25.99, *p* = 0.245 vs. wt-DH) (Figure 7E,F, inset). In functional terms, it is interesting to note that the increase in eIPSC amplitude and the alleviation of frequency-dependent depression in dorsal CA1 pyramidal cells of dnActRIB mice do not translate into higher synaptic efficacy i.e., enhanced charge transfer, because, at the same time, eIPSC decay is significantly accelerated. A possible interpretation would be that activin signaling is predominantly tailored to regulate the temporal window of fast synchronized inhibitory events rather than their overall efficacy.

### 2.7. Activin Signaling Is Critically Involved in the Effects of EE on eIPSCs

To complete our study, we asked how EE affects eIPSCs and examined the role of activin signaling therein. In ventral CA1 cells from wt mice, EE housing did not lead to a significant shift of the control I-O curve (n = 19, two-way repeated measures ANOVA, F = 2.69, *p* = 0.110; Figure 8A). By contrast, the I-O curve of dorsal CA1 cells exhibited a pronounced upward shift after EE (n = 14, two-way repeated measures ANOVA, F = 6.56, *p* = 0.020; Figure 8B). As a consequence of its dorsal-only action, EE leveled out the imbalance in the I-O relationship between ventral and dorsal pyramidal cells that prevailed in the absence of behavioral stimulation (response to 200 µA stimulation intensity, wt-VH-EE: n = 19, 2.38 ± 0.22 nA; wt-DH-EE: n = 14, 2.97 ± 0.34 nA, *p* = 0.134; Figure 8A,B). The dorsal–ventral disparity of eIPSC rise time in the control hippocampus was also equalized after EE (wt-VH-EE: n = 19, 2.93 ± 0.28 ms; wt-DH-EE: n = 14, 3.75 ± 0.40 ms, *p* = 0.096), whereas the difference in decay tau between the two hippocampal regions persisted in the enrichment group (wt-VH-EE: n = 19, 43.22 ± 3.35 ms, wt-DH-EE: n = 14, 58.56 ± 4.78 ms, *p* = 0.010) (Table 2). By selectively enhancing dorsal eIPSC amplitude without affecting its slow decay, EE introduced a bias in synaptic efficacy in favor of dorsal inhibition that was absent in hippocampi from mice in control housing. In numbers, the total charge transfer, which matched in control cells between DH and VH, increased after EE in dorsal cells to 0.20 ± 0.02 nA*ms (n = 14), which was significantly higher than in ventral cells after EE (0.12 ± 0.02 nA*ms, n = 19, *p* = 0.006, Figure 8A,B; Table 2).

Regarding the behavior of synaptic responses during repetitive electrical stimulation, EE promoted frequency-dependent depression in ventral neurons (n = 14; two-way repeated measures ANOVA, F = 9.15, *p* = 0.005; Figure 8C), while this parameter remained unaffected by EE in dorsal cells (Figure 8D). When calculated exemplarily for the fifth stimulus in ventral recordings, the relative eIPSC amplitudes (normalized to first stimulus) amounted to 0.63 ± 0.15 (n = 19) before and to 0.52 ± 0.12 after EE (n = 14, *p* = 0.030). As a consequence of enhanced frequency depression in ventral hippocampus, significantly less charge was transferred during the train (wt-VH-EE: 306.28 ± 19.22 nA*ms, *p* = 0.034 vs. wt-VH, Figure 8C). EE housing did not alter frequency depression in dorsal cells (n = 14, F = 1.08, *p* = 0.309; Figure 8D) nor did it affect normalized charge transfer (404.04 ± 30.93 nA*ms, *p* = 0.320 vs. wt-DH, Figure 8D), which then became significant compared to VH (*p* = 0.012, wt-VH-EE vs. wt-DH-EE).

Finally, we probed eIPSC properties in hippocampi from dnActRIB mice with and without EE housing. Loss of activin signaling clearly abrogated the prominent effects of EE in wt cells, namely the upward shift of the dorsal I-O relationship (Figure 8F; effect of EE on dorsal dnActRIB cells. n = 8, two-way repeated measures ANOVA, F = 0.27, *p* = 0.608) and the enhanced frequency depression in VH (Figure 8G; effect of EE on ventral dnActRIB cells. n = 18, two-way repeated measures ANOVA, F = 1.81, *p* = 0.188; relative charge transfer in ventral dnActRIB EE cells, 345.26 ± 30.05, *p* = 0.52 vs. dnActRIB-con cells).

## 3. Discussion

### 3.1. Hippocampus Exhibits Regional Differences in GABAergic Inhibition

In the present study, we systematically characterized and compared the various features of GABA_A_R-mediated inhibition of CA1 pyramidal cells from the dorsal and ventral hippocampus of adult mice. Furthermore, we elucidated the role of activin signaling in shaping regional differences in inhibition.

CA1 pyramidal cells from the ventral hippocampus exhibited larger amplitudes of tonic GABA current than cells from the dorsal part. Given that the α5 subunit is a characteristic constituent of extrasynaptic GABA_A_Rs in CA1 pyramidal cells [40], the stronger tonic current in ventral neurons is consistent with the higher density of α5 subunits in ventral compared to dorsal CA1 neurons [36]. While spontaneous IPSCs rose faster in ventral than in dorsal CA1 neurons, they did not exhibit significant regional differences with respect to frequency, amplitude, and decay time constant. These findings disagree in part with earlier studies, which reported a higher frequency of miniature IPSCs in VH than DH [29], or observed equal frequencies, but larger amplitudes, of spontaneous events in dorsal vs. ventral neurons [41]. The apparent discrepancies to our data might be explained on the grounds that the abovementioned studies were performed at room temperature [29] and/or on immature hippocampal neurons from a different species (2–3-week-old rats) [41]. Furthermore, differences in the angle at which the hippocampus was sliced might have possibly affected the local pattern of synaptic connectivity intrinsic to each slice preparation. Whereas we sectioned the brain along the horizontal plate to obtain slices containing the hippocampus at different dorsoventral levels, in the other mouse study referenced above [29], the hippocampus was isolated and its bent structure straightened first, before slices were cut perpendicular to the longitudinal axis. The two cutting procedures should have yielded hippocampal slices with different degrees of transversality relative to the length axis, but it seems rather unlikely that this methodological variation alone should have given rise to major discrepancies in IPSC properties between the preparations.

Electrically evoked IPSCs had larger amplitudes in ventral than in dorsal CA1 neurons. However, eIPSCs did not differ between regions, if their inhibitory potency was described in terms of total charge transfer, because the ventral events displayed a significantly faster decay. In apparent contrast, Maggio and Segal [41], recording from immature rat neurons (v.s.), reported a larger amplitude of evoked events in dorsal compared to ventral neurons, with no difference in kinetics. In another study on adult rat pyramidal cells, electrically evoked inhibitory postsynaptic potentials (rather than currents) were larger but also broader in dorsal than in ventral neurons [42]. From our results, we would propose that, owing to their larger peak amplitude in combination with faster decay kinetics, evoked events in mouse ventral CA1 pyramidal cells appear to be tuned to effectuate a more potent, but also temporally more confined, window of phasic inhibition than in dorsal cells. Upon repetitive stimulation, dorsal neurons exhibited more pronounced frequency-dependent depression of eIPSCs than ventral ones, but, again, total charge transfer during stimulus trains was about equal between cells from DH and VH due to the slower decay kinetics in dorsal neurons.

Why does eIPSC deactivation appear to be slower in dorsal than in ventral neurons? Although DH and VH are endowed with distinct expression profiles of GABA_A_R subunits [36], a differential subunit composition of GABA_A_Rs in dorsal and ventral pyramidal cells is unlikely to account for the altered decay kinetics of electrically evoked events, because spIPSCs did not exhibit an appreciable difference in current deactivation. The fact that both dorsal and ventral eIPSCs decayed with a considerably slower time course than spontaneous and miniature events suggests two alternative explanations, which are not mutually exclusive. Firstly, electrical stimulation in the dorsal CA1 stratum radiatum might elicit less synchronized transients of vesicular GABA release than in the ventral portion, possibly reflecting variations in the timing of interneuron recruitment and/or in the properties of the interneurons, including the routing of their axonal projections to pyramidal cells. In any case, variations in the degree of synchrony of evoked GABA release would be expected to affect eIPSC peak amplitude and kinetics [43]. Secondly, the amount of GABA released upon electrical stimulation might overload the capacity of glial and neural GABA transporters in a region-specific fashion, so that removal of the transmitter from the synaptic cleft is delayed in DH relative to VH. Importantly, the remaining GABA concentration not yet cleared from the extracellular space may attenuate the amplitude of subsequent synaptic responses during repetitive stimulation [44], due to accumulation of receptors in a slow desensitized state [45]. The stronger frequency-dependent depression of eIPSCs that we observed in dorsal neurons lends support to the idea that GABA reuptake may be less efficient in DH compared to VH. This would also be consistent with the above notion that ventral uptake mechanisms should be stronger than dorsal ones to curtail tonic inhibition in ventral CA1 cells.

### 3.2. Effect of Activin in Dorsal and Ventral Hippocampus

Whereas type I and type II activin receptors are distributed widely across the hippocampus [8], we report here that the basal level of activin A exhibits a pronounced dorsal–ventral gradient, with a fourfold decline from the dorsal to ventral pole. As almost predicted by the low level of activin A in VH, genetic disruption of activin signaling did not affect any feature of tonic nor phasic inhibition in ventral neurons. By contrast, loss of activin signaling increased dorsal GABAergic tonic current and spIPSC frequency, enhanced the amplitude of dorsal eIPSCs while accelerating their decay, and reduced frequency-dependent depression of subsequent eIPSCs. Thus, significant dorsal–ventral differences in wt hippocampi were equalized in the mutant preparation. In other words, functional activin signaling contributes to the dorsal–ventral disparity of some essential properties of phasic inhibition.

EE is a potent behavioral strategy to enhance cognition, stabilize emotional imbalance, and protect from mental disorders [46]. At the neuronal level, EE has been shown to modulate cellular excitability, synaptic plasticity, and network activity [10,47,48]. Because overnight EE exposure engenders a transient surge of activin A throughout all regions of the hippocampus (Figure 1), we used this behavioral paradigm in conjunction with dnActRIB mice to determine to what extent activin signaling is involved in the effects of EE on GABA_A_R-mediated inhibition. EE produced a prominent reduction of GABAergic tonic current and spIPSC frequency selectively in ventral CA1 neurons. Since these effects were absent in ventral hippocampi from dnActRIB mice and reproduced by recombinant activin A in ventral wt slices, the reduction in ventral spIPSC frequency can be best explained as a consequence of the EE-induced rise in activin A. In a similar vein, activin signaling appears to be responsible for the augmented frequency depression of ventral eIPSCs after EE, as this effect was abrogated in ventral neurons from dnActRIB mice exposed to EE. These findings strongly suggest that the EE-evoked surge in activin A serves to adjust the neurophysiological phenotype in response to a preceding behavioral experience. This is not to say that other genes whose transcription is also upregulated by EE do not play a role in this context. For example, *brain-derived neurotrophic factor* (BDNF), which is also responsive to EE [12,49,50], has been shown to modulate hippocampal GABA function as well, leading to an increase in mIPSC frequency and an enhancement of eIPSC amplitude [51].

A transcriptional analysis performed 1 h after a very brief period (15 min) of novel environment exposure identified 39 differentially expressed genes between CA1 neurons activated by this behavioral paradigm and those not activated [50]. Among them was *Bdnf*, but not *Inhba*, the gene encoding the βA subunit of homodimeric activin A protein (βAβA). Regarding the unexpected lack of *Inhba* upregulation, one might suspect that the novel environment exposure of 15 min in that study was simply too short, compared to the 12 h EE regimen in our study, to obtain a significant upregulation of *Inhba* expression. However, when the same transcriptional comparison was made between activated and non-activated dentate gyrus granule neurons, *Inhba* was at the very top of 749 differentially expressed genes, second only to *Arc* [50]. Thus, the short exposure period alone is unlikely to account for the absent *Inhba* upregulation in activated CA1 neurons. Rather, the high baseline activity of CA1 neurons proved to be a mitigating factor in the overall transcriptional response to novelty exploration compared to the sparsely active dentate granule neurons, which exhibited a huge transcriptional response, once activated by novelty exposure [50]. Since we have firmly established an appreciable upregulation of *Inhba* mRNA in CA1 [12], followed by a strong rise in activin A protein after 12 h of EE (Figure 1), it seems likely that the spectrum of transcribed genes in CA1 becomes broader as mice are allowed more time to explore a novel and stimulating environment. Nevertheless, our previous in situ hybridization of hippocampal *Inhba* demonstrated that even after 12 h of EE exposure, the striking discrepancy of mRNA signals between dentate gyrus and CA1, observed after very short novelty exploration [50], was not fully resolved, as there was still a considerable gradient in signal strength preserved, manifested as a decline along the canonical trisynaptic circuit of the hippocampus [12].

Thus, activin signaling in the hippocampus appears to be segregated in two spatial dimensions. In addition to the dorsal–ventral gradient along the longitudinal axis of the hippocampus, which was the focus of the present study, the transverse axis exhibits a dentate gyrus-to-CA1 gradient. It is tempting to hypothesize that the area-specific expression of the activin system and its differential responsiveness to stimulating activity serves to adapt the cellular and molecular machinery in the various hippocampal regions to their particular needs in cognitive function and affective behavior.

## 4. Materials and Methods

### 4.1. Animals

Male and female adult mice with a C57Bl/6J background (2.5–5 months old) and transgenic mice expressing dnActRIB under the control of the CaMKIIα promoter [7] were used for experiments. Animals were housed under standard conditions. All procedures were conducted according to the guidelines of the Animal Protection Law of Germany and the European Communities Council Directive of 24 November 1986 (1986/86/609/EEC), with approval of local government of Lower Franconia.

### 4.2. Enriched Environment

EE exposure was performed as described previously [12]. Briefly, pairs of mice were placed in large cages (60 cm × 38 cm × 18 cm), which were equipped with shelters, toys, and tunnels for 12 h from 8 p.m. to 8 a.m., during their dark phase.

### 4.3. Activin Enzyme-Linked Immunosorbent Assay (ELISA)

Mice were sacrificed under sevoflurane or isoflurane anesthesia, and brains were rapidly dissected out from control mice and mice after overnight EE housing. Dorsal, medial, and ventral regions were cut from the isolated hippocampi and mechanically broken down with a magnetic homogenizing ball for 30 s in lysis buffer containing 0.32 M sucrose, 5 mM Tris-HCl (pH 8.0), and protease inhibitor cocktail (Sigma-Aldrich, Steinheim, Germany), and homogenates were centrifuged at 13,000× *g* at 4 °C for 10 min (twice). Supernatants were collected and activin A levels were assayed by an ELISA kit (R&D systems, via Bio-techne GmbH, Wiesbaden, Germany) according to the manufacturer’s instructions.

### 4.4. Slice Preparation and Electrophysiological Recordings

Mice were anesthetized with sevoflurane or isoflurane and decapitated. The brains were quickly removed and trimmed with a cut between cerebellum and cerebral cortex. Horizontal sections (350 µm thick) from each hemisphere were cut to obtain hippocampal slices at different dorsoventral levels, in ice-cold sucrose-based artificial cerebrospinal fluid (aCSF) containing (in mM) 75 sucrose, 87 NaCl, 3 KCl, 0.5 CaCl_2_, 7 MgCl_2_, 1.25 NaH_2_PO_4_, 25 NaHCO_3_, and 10 d-glucose. The first two and the last two slices per hemisphere containing the complete hippocampal formation from the dorsal or ventral pole were collected and incubated in the same solution for 10 min at 35 °C, and then allowed to rest in aCSF containing (in mM) 125 NaCl, 3 KCl, 1 CaCl_2_, 3 MgCl_2_, 1.25 NaH_2_PO_4_, 25 NaHCO_3_, and 10 d-glucose at room temperature for at least 2 h before being used. Individual slices were transferred to the submerged recording chamber, which was perfused by means of a gravity-driven system with standard aCSF containing 1.5 mM MgCl_2_ and 2.5 mM CaCl_2_ at 31 ± 1 °C. All solutions were constantly gassed with 95% O_2_–5% CO_2_.

Whole-cell recordings of visualized CA1 pyramidal cells were performed with patch pipettes filled with (in mM) 130 CsCl, 3 MgCl_2_, 5 ethylene glycol tetraacetic acid (EGTA), 5 4-(2-hydroxyethyl)-1-piperazineethanesulfonic acid (HEPES), 2 Na_2_-ATP, 0.3 Na-GTP, 5 QX-314 (pH 7.3). Electrode resistance with internal solution ranged from 3 to 5 MΩ. In whole-cell voltage-clamp mode, series resistance was about 8–20 MΩ and compensated by 60–80%. All recordings were performed at −70 mV (after correction for liquid junction potential) and in the presence of kynurenic acid (2 mM) in the extracellular solution to block glutamatergic synaptic transmission. Signals were filtered at 2 kHz and sampled at 20 kHz using a Multiclamp 700B amplifier in conjunction with Digidata 1440A interface and pClamp10.2 software (Molecular Devices, Sunnyvale, CA, USA).

GABAergic tonic inhibition was calculated from the shift of holding current in response to the GABA_A_R blocker picrotoxin (100 µM), in the presence of a low concentration of GABA (5 µM) to maintain a constant level of ambient GABA in the slices [40]. GABAergic phasic inhibition was investigated by monitoring the spontaneously occurring IPSCs, in the absence or presence of TTX (0.5 µM; to block action potentials) for spontaneous IPSCs and miniature IPSCs, respectively. In some experiments, recombinant activin A (25 ng/mL; R & D system) was added to the perfusion solution for 5–10 min to monitor its acute effects on such phasic events. The recording was continued for at least 10 min after drug wash-out to demonstrate recovery from drug effects. To obtain electrically evoked IPSCs in CA1 pyramidal cells, a concentric platinum bipolar electrode was placed in the stratum radiatum close to the pyramidal cell layer. Constant current pulses were delivered at 0.1 ms width, with an inter-stimulus interval of more than 1 s. Input–output relationships for evoked IPSCs in dorsal and ventral CA1 pyramidal cells were determined by plotting peak current amplitude as a function of stimulus strength, ranging from 50 to 200 µA. Frequency-dependent depression of evoked IPSCs was gauged using trains of 15 stimuli at 5 Hz (inter-pulse interval of 200 ms), with stimulus intensity set to obtain 60–80% of maximal amplitude for the first current response.

### 4.5. Data Analysis

Data analysis was performed off-line with Clampfit 10.6 (Molecular Devices, Sunnyvale, CA, USA). Spontaneous and evoked inhibitory events were analyzed as previously described [20]. Briefly, fast spontaneous events were identified using an automated detection algorithm over at least 2 min long recordings. These events were then averaged to measure amplitude and kinetics including 10–90% rise time and decay time constant. Frequency-dependent depression of evoked IPSCs was determined by normalizing the peak amplitude of subsequent responses to that of the first response in the train. OriginPro 2018G (OriginLab Corporation, Northampton, MA, USA) was used for statistics and figures. Data were expressed as means ± SEM. Statistical comparisons of data were performed using two-way repeated measures ANOVA or Student’s *t*-test as appropriate. Significance was assumed for *p* < 0.05.

## Figures and Tables

**Figure 1 ijms-24-13145-f001:**
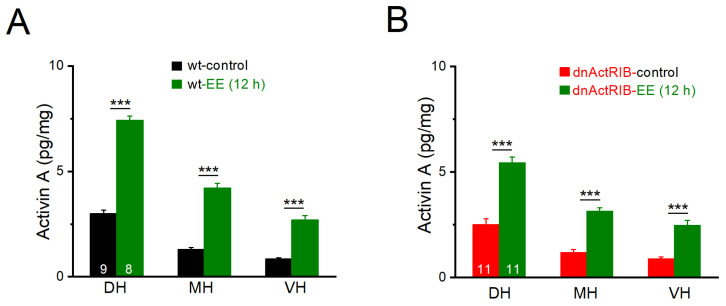
Dorsal–ventral gradient of activin A before and after enriched environment (EE) exposure for 12 h. Activin A level in dorsal, medial, and ventral hippocampus (DH, MH, and VH) from wt (**A**) and dnActRIB (**B**) mice was determined by ELISA. Numbers in columns indicate sample size. *** *p* < 0.001.

**Figure 2 ijms-24-13145-f002:**
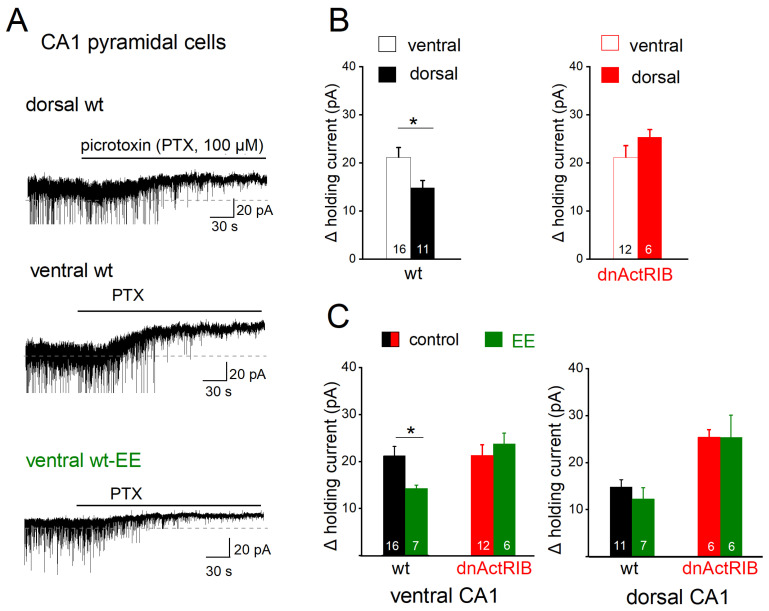
GABA_A_ receptor-mediated tonic inhibition in dorsal and ventral regions is differently modulated by activin signaling and EE. All recordings were performed in voltage-clamp mode (V_h_ −70 mV), with GABA (5 µM) and ionotropic glutamate receptor blocker kynurenic acid (KA, 2 mM) in bathing solution. (**A**) Representative traces of wt CA1 pyramidal cells without and with EE housing illustrate current shift in response to GABA_A_R blocker picrotoxin (PTX, 100 µM). Note that tonic current and spontaneous phasic events (partially truncated) are downward due to high Cl^−^ concentration in pipette solution. (**B**) Comparison of changes in holding current in dorsal and ventral regions of wt mice (left) reveals significant larger PTX-sensitive current in ventral cells. Such dorsal–ventral difference in tonic inhibition was abrogated in dnActRIB slices, due to a selective enhancement dorsally. (**C**) EE housing reduced tonic inhibition only ventrally in wt mice. * *p* < 0.05.

**Figure 3 ijms-24-13145-f003:**
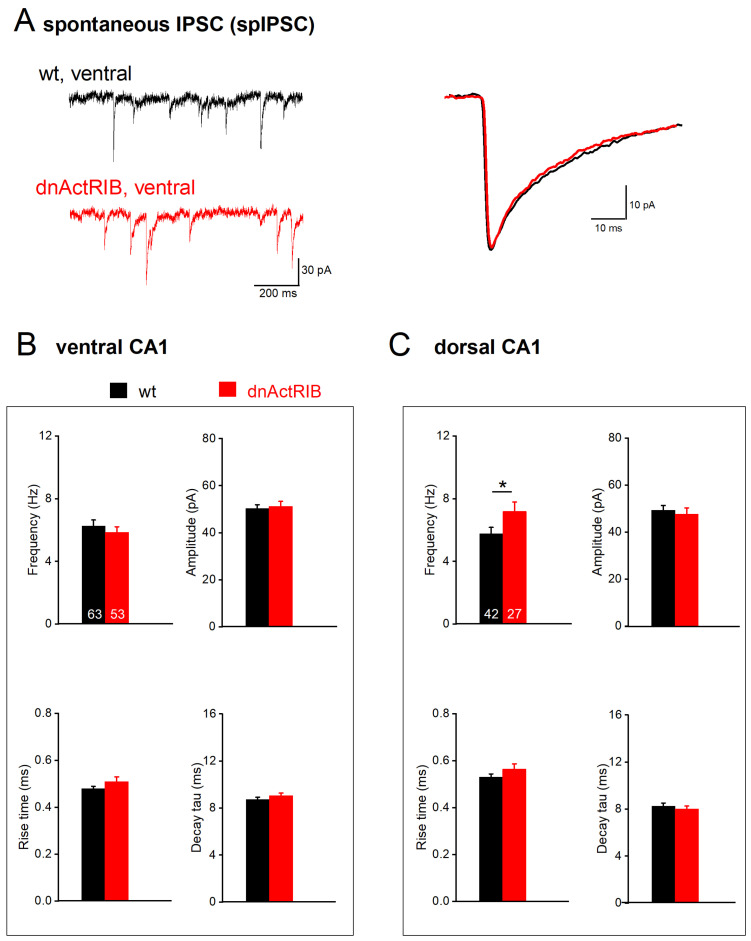
Disruption of activin signaling enhances spIPSC frequency exclusively in dorsal CA1 pyramidal cells. (**A**) Typical current traces from ventral pyramidal cells of wt (black) and dnActRIB (red) animals show spIPSCs in the presence of KA. Superimposed traces on the right are like-colored averaged events from corresponding cells on the left. (**B**,**C**) Histograms summarize impact of activin signaling on spIPSCs in ventral (**B**) and dorsal (**C**) hippocampus, with respect to frequency (top left), peak amplitude (top right), and kinetics (10–90% rise time, bottom left; and decay tau, bottom right). * *p* < 0.05.

**Figure 4 ijms-24-13145-f004:**
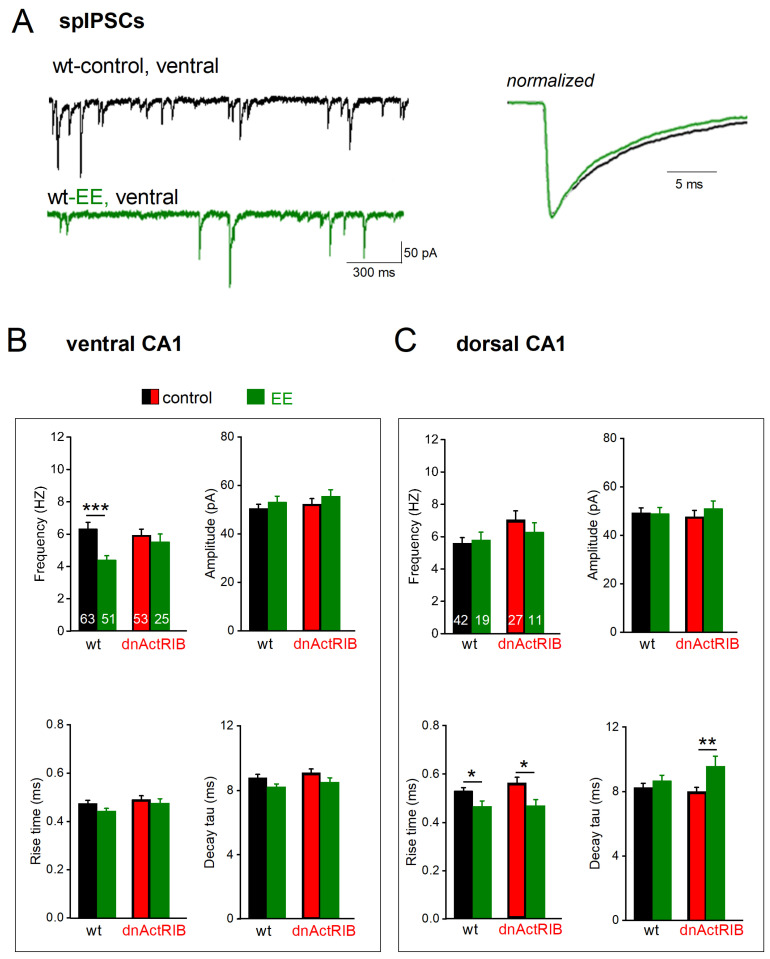
EE-induced surge in activin A decreases spIPSC frequency in ventral CA1 pyramidal cells. (**A**) Representative current recordings from ventral CA1 neurons from a wt mouse before (black) and a wt mouse after 12 h EE (green, left-hand side). Averaged spIPSCs from cells on the left were normalized to peak amplitude and superimposed to compare kinetics (right-hand side). (**B**,**C**) Histograms summarize impact of EE (green) on frequency (top left), amplitude (top right), and kinetics (bottom) in both wt (black) and dnActRIB (red) mice in ventral (**B**) and dorsal (**C**) hippocampus. * *p* < 0.05, ** *p* < 0.01, *** *p* < 0.001.

**Figure 5 ijms-24-13145-f005:**
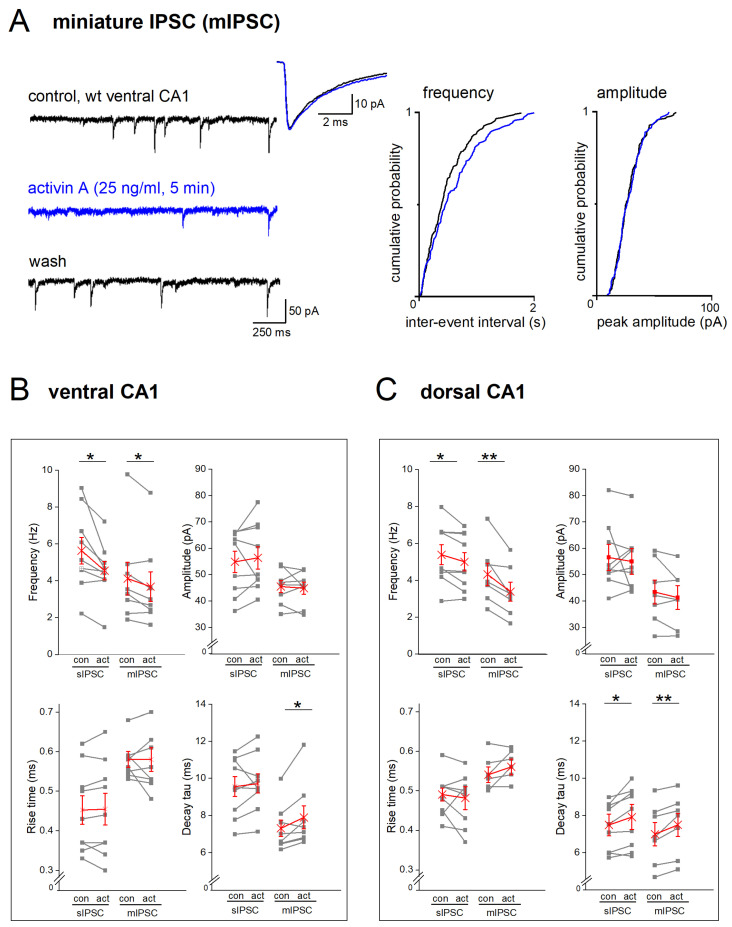
Recombinant activin A reduces frequency of spontaneous and miniature IPSCs in dorsal and ventral CA1 pyramidal cells from wt mice. (**A**) Reversible inhibition by activin A (25 ng/mL for 5 min) of miniature events (in TTX) in a ventral neuron. Raw traces on the left were obtained before (control) and during (blue) activin A superfusion and 6 min after wash. Inset above depicts superimposition of averaged mIPSCs over 2 min recording period before (black) and during maximal effect of activin application (blue). Plots on the right show the cumulative probability for inter-event interval and peak amplitude of the corresponding mIPSCs. (**B**,**C**) Summary of activin effects on spIPSC (left) and mIPSC properties (right), including frequency (top left), averaged peak amplitude (top right), rise time (bottom left), and decay tau (bottom right) in cells from both ventral (**B**) and dorsal (**C**) hippocampi. Single data points are displayed in gray, average in red. con: control; act: activin A. * *p* < 0.05; ** *p* < 0.01.

**Figure 6 ijms-24-13145-f006:**
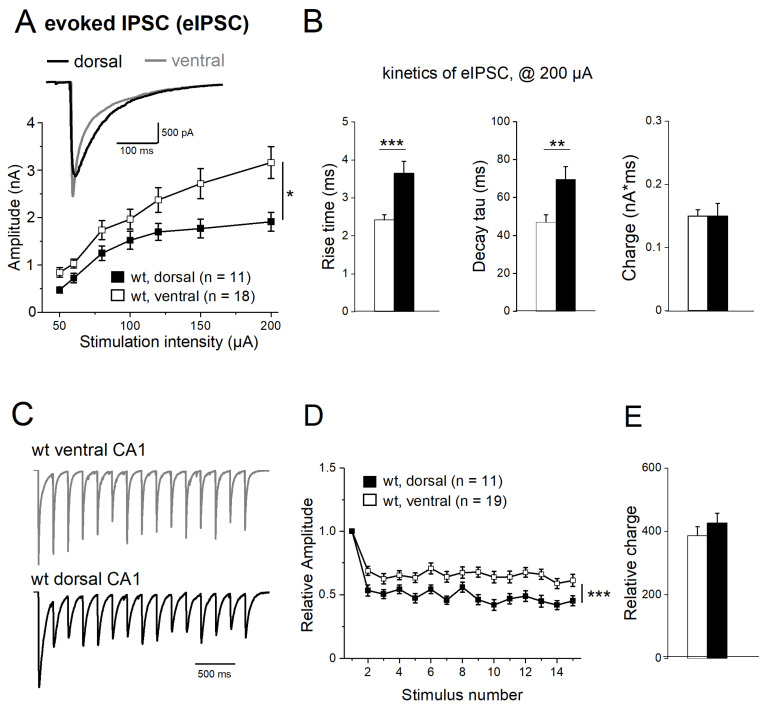
Dorsal–ventral difference of electrically evoked IPSCs (eIPSCs). (**A**) Input–output curves depict eIPSC peak amplitude as a function of stimulus intensity. Inset shows superimposed eIPSC response to a 200 µA stimulus in dorsal (black) and ventral CA1 cell (gray). (**B**) Histograms summarize eIPSC kinetics and charge transfer in response to 200 µA stimuli. (**C**,**D**) Stimulus trains (100 µA, 5 Hz) reveal stronger frequency-dependent depression of normalized eIPSCs in dorsal than ventral CA1 neurons (**C**), which is graphically summarized in (**D**). (**E**) Histogram of total charge transfer during stimulus train. * *p* < 0.05, ** *p* < 0.01, *** *p* < 0.001.

**Figure 7 ijms-24-13145-f007:**
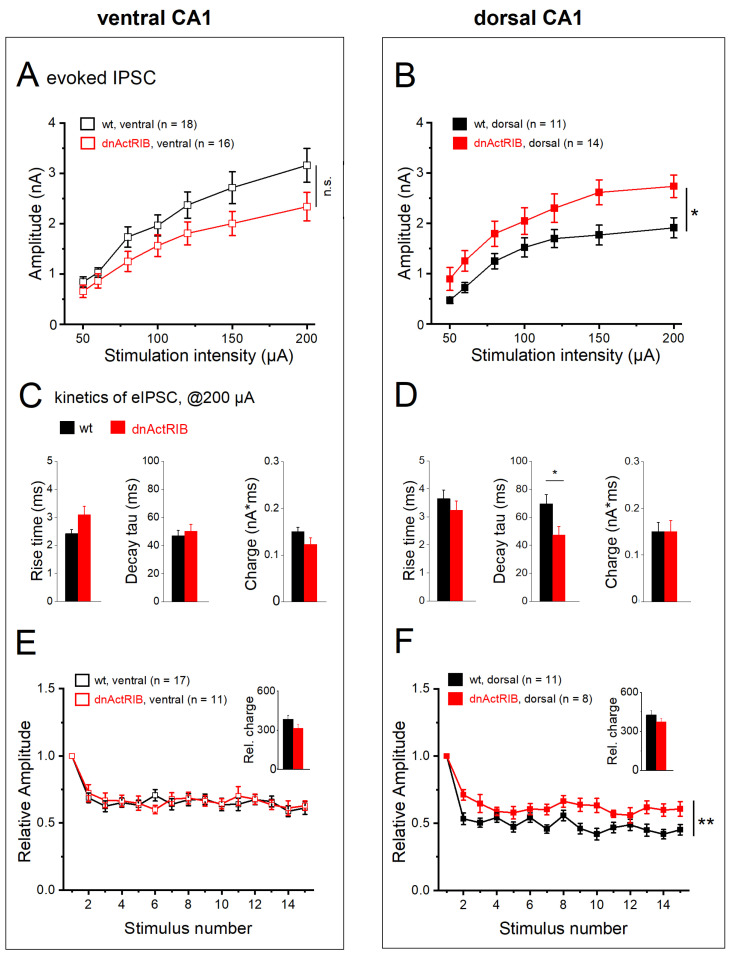
Activin signaling-deprived CA1 neurons from dorsal hippocampus exhibit enhanced eIPSC response. (**A**,**B**) Input–output curves for eIPSCs in ventral (**A**) and dorsal (**B**) CA1 pyramidal cells from wt and dnActRIB mice, respectively. (**C**,**D**) Histograms summarize properties of eIPSC elicited at 200 µA in ventral (**C**) and dorsal hippocampus (**D**) from wt and mutant mice. (**E**,**F**) Mutant cells from dorsal, but not ventral, hippocampus show attenuated frequency-dependent depression during train. Overall charge transfer was computed from normalized eIPSC waveforms (insets). n.s., not significant. * *p* < 0.05, ** *p* < 0.01.

**Figure 8 ijms-24-13145-f008:**
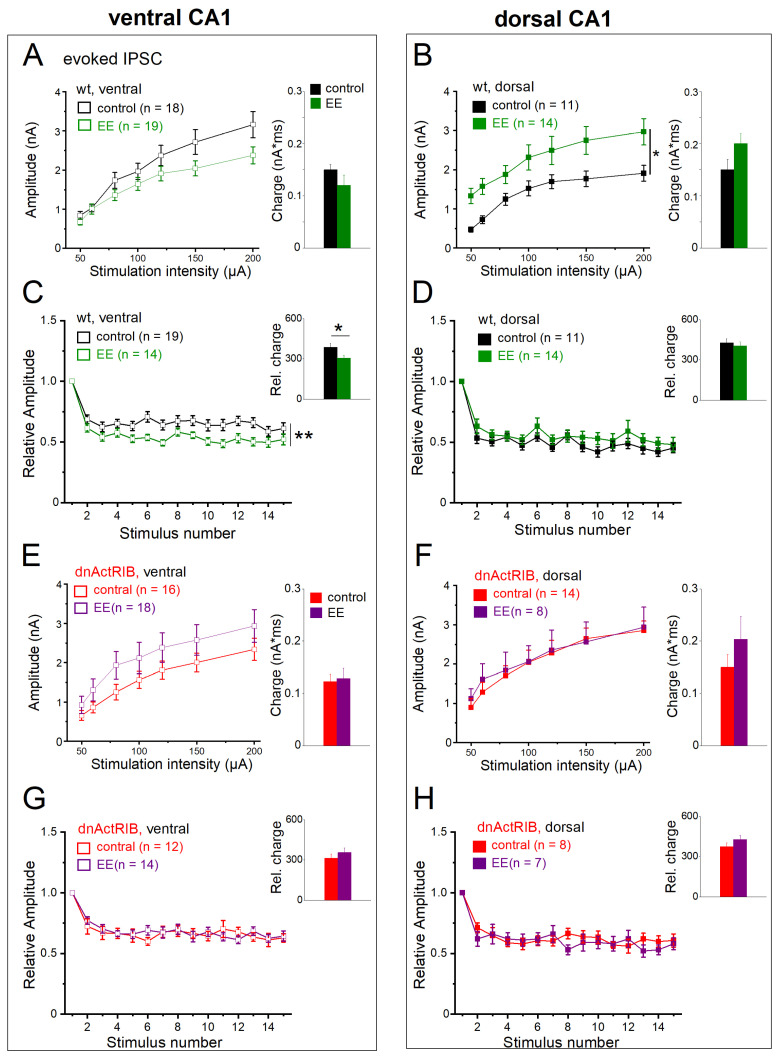
Effects of EE on single and repetitive eIPSCs in wt and mutant CA1 neurons from dorsal and ventral hippocampus. (**A**,**B**) Input–output curves (left) and charge transfer for single eIPSCs at 200 µA stimulus strength (right) in ventral and dorsal CA1 neurons from control and EE-exposed wt mice. (**C**,**D**) EE enhanced frequency-dependent depression and reduced charge transfer (inset) in wt ventral CA1 cells (**C**), whereas their dorsal counterparts remained unaffected (**D**). (**E**–**H**) In dnActRIB hippocampi, the effects of EE on input–output relationship in dorsal hippocampus and frequency-dependent depression and charge transfer in ventral hippocampus were both abrogated. * *p* < 0.05, ** *p* < 0.01.

**Table 1 ijms-24-13145-t001:** Properties of spIPSCs in dorsal and ventral CA1 pyramidal cells.

spIPSCs		Frequency (Hz)	Amplitude (pA)	Rise Time (ms)	Decay Tau (ms)	Half-Width (ms)	Charge (pA*ms)
**wt-con**	ventral (63)	6.26 ± 0.38	50.27 ± 1.63	0.48 ± 0.01 **	8.72 ± 0.21	6.03 ± 0.16	466.09 ± 17.37
	dorsal (42)	5.77 ± 0.40	49.35 ± 2.00	0.53 ± 0.01	8.26 ± 0.26	5.73 ± 0.18	457.19 ± 16.79
**dnActRIB-con**	ventral (53)	5.86 ± 0.35 *	51.19 ± 2.14	0.51 ± 0.02	9.05 ± 0.23 **	6.32 ± 0.22	515.19 ± 16.79
	dorsal (27)	7.19 ± 0.60 #	47.74 ± 2.58	0.56 ± 0.02	8.02 ± 0.24	5.68 ± 0.21	447.53 ± 24.78
**wt-EE**	ventral (51)	4.41 ± 0.27 *,§§§	53.06 ± 2.37	0.45 ± 0.01	8.17 ± 0.19	5.47 ± 0.14 §	489.68 ± 238.51
	dorsal (19)	5.81 ± 0.47	49.02 ± 2.52	0.47 ± 0.02 §	8.58 ± 0.34	5.82 ± 0.25	454.92 ± 23.13
**dnActRIB-EE**	ventral (25)	5.53 ± 0.48	55.67 ± 2.62	0.48 ± 0.02	8.56 ± 0.28	6.07 ± 0.25	506.33 ± 27.64
	dorsal (11)	6.28 ± 0.57	51.15 ± 3.07	0.47 ± 0.02 §	9.59 ± 0.61 §§	6.30 ± 0.50	502.98 ± 13.15

Data are expressed as mean ± SEM. Ventral values are displayed over gray background. Statistical comparison: * (dorsal vs. ventral); # (wt vs. dnActRIB); § (con vs. EE). * *p* < 0.05, ** *p* < 0.01, (same significance levels apply to §, §§, §§§ and #).

**Table 2 ijms-24-13145-t002:** Kinetics of evoked IPSCs in dorsal and ventral CA1 pyramidal cells.

eIPSCs		Amplitude (pA)	Rise Time (10–90%; ms)	Decay Tau (ms)	Half-Width (ms)	Charge (nA*ms)
**wt-con**	ventral (18)	3.16 ± 0.33 *	2.42 ± 0.14 ***	47.12 ± 0.37 **	28.67 ± 2.71 **	0.15 ± 0.01
	dorsal (11)	1.91 ± 0.20	3.65 ± 0.31	69.61 ± 6.83	48.95 ± 5.75	0.15 ± 0.02
**dnActRIB-con**	ventral (16)	2.34 ± 0.28	3.09 ± 0.31	50.33 ± 4.85	32.44 ± 3.93	0.12 ± 0.01
	dorsal (14)	2.73 ± 0.22 #	3.24 ± 0.33	47.29 ± 6.11 #	29.36 ± 4.48 #	0.15 ± 0.02
**wt-EE**	ventral (19)	2.38 ± 0.22	2.93 ± 0.28	43.22 ± 3.35 **	29.06 ± 2.64 **	0.12 ± 0.01 **
	dorsal (14)	2.97 ± 0.34 §	3.75 ± 0.40	58.56 ± 4.73	44.03 ± 4.34	0.20 ± 0.02
**dnActRIB-EE**	ventral (18)	2.94 ± 0.42	2.83 ± 0.26	42.44 ± 4.23 *	28.10 ± 3.59	0.13 ± 0.02
	dorsal (8)	2.94 ± 0.51	3.28 ± 0.34	56.68 ± 4.67	38.67 ± 2.69	0.20 ± 0.04

IPSCs were evoked by a 200 µA stimulus. Ventral values are displayed over gray background. Statistical comparison: * (dorsal vs. ventral cells); # (wt vs. dnActRIB); § (con vs. EE). * *p* < 0.05, ** *p* < 0.01, *** *p* < 0.001 (same significance levels apply to § and #).

## Data Availability

Data of this study are available from the corresponding author upon reasonable request.

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
