# Peer review of "Dorsal–Ventral Gradient of Activin Regulates Strength of GABAergic Inhibition along Longitudinal Axis of Mouse Hippocampus in an Activity-Dependent Fashion"

_ijms, 2023, doi:10.3390/ijms241713145_

Round 1
Reviewer 1 Report
The authors present the design, realization and interpretation of scientific results with meritorious rigor. Some sources of errors are identified (such as the difference between species or the maturation of the nervous structures) able to explain in detail the apparent discrepancies compared to the specialized literature. However, the authors avoid giving a clinical impact to their own results, although this is obvious, they avoid discussing the gradients identified in the context of signaling influenced by macroscopic structures (which influence the diffusion of mediators among others). If the authors put more effort into identifying the applications of their own results, their own impact through other researchers who cite them can be more important than the decrease of their own subsequent developments (in the absence of competition from readers). For example, the authors could extrapolate the data regarding the differences between the genetic model and the behavioral one, using the signaling pathways and structures involved (the simplest variables being the shape, connectivity, permeability of mediators through the cellular matrix... but not the most interesting ones).
Author Response
We thank the reviewer for the appreciation of the scientific merit of our manuscript. We are also grateful to the reviewer for suggesting a different publication strategy, focusing more on translational and clinical aspects of our findings. In the present paper, we laid the grounds for understanding how the activin gradient shapes the dorsal-ventral difference in GABAergic neurotransmission in the hippocampus under physiological conditions. We didn´t extend our experiments on animal models of neuropsychiatric disorders. Thus, we refrained from emphasizing putative clinical implications at the current stage of our activin studies, but we will certainly keep this important aspect in mind when exploring the role of activin receptor signaling in the context of disease models.
Reviewer 2 Report
The manuscript by Valero-Aracama and coworkers shows a dorsal-ventral gradient of GABAergic inhibition in the mouse hippocampus. The authors convincingly demonstrate that the gradient is caused by the differential expression of the growth factor actinin. Analysis is done with a range of electrophysiological, behavioral and molecular methods, all of which were applied in a competent and well-controlled way. The findings add to numerous doral-ventral grandients in the hippocampus at the molecular, cellular and network level. The present results are particularly interesting as they involve an important behavioral paradigm (enriched environment) and trace down the observations to a causal molecular event (expression of a growth factor).
I have four comments/concerns, all meant in a constructive way:
1. All electrophysiological experiments were made in horizontal brain slices. As a consequence, cutting angles relative to the longitudinal hippocampal axis vary between ventral-, mid- and dorsal hippocampal slices. Can this possibily affect connectivity and, hence, the properties of spontaneous or evoked inhibition? A control experiment with sections perpendicular to the length axis would be ideal, but at least the point should be discussed.
2. Amplitudes are reported as means of peak current amplitudes. Given the highly non-normal distribution of synaptic current amplitudes, a better description are (cumulative) amplitude histograms which are then tested for differences, e.g. using the Kolmogoror-Smirnov-Test. Mean values may not be as sensitive. In the present work, most differences were in frequency of synaptic currents, rather than amplitude. However, a shift in amplitude distribution may result in an unchanged mean amplitude value, but still a larger fraction of currents below the detection threshold. Based on the consistent findings I dont't think this is the case, but nevertheless it should be controlled by -at least an exemplary- distribution analysis.
3. The time course of action af bath-applied activin (Fig. 5) should be documented and commented. This may shed some light on the mechanism which is expected to be rather indirect. Was activin dissolved in water/ACSF? Was there any control experiment with other peptides?
4. The discussion contains relatively little information about the proposed cellular and molecular mechanism.
The manuscript should be once more proof-read for typos and minor grammatical errors. Examples (all from the abstract):
Line 2: his ==> this
Line 14: activing ==> activating
